

**First GPS TEC maps of ionospheric disturbances induced**
**by reflected tsunami waves: The Tohoku case study**
**L. Tang[1, 2], Y. Zhao[1] and J. An[3]**
[1] Key Laboratory of Earthquake Geodesy, Institute of Seismology, China Earthquake
Administration, Wuhan 430071, China
[2] School of Geodesy and Geomatics, Wuhan University, Wuhan 430079, China
[3] Chinese Antarctic Center of Surveying and Mapping, Wuhan University, Wuhan 430079,
China
*Correspondence to*: L. Tang (ltang@whu.edu.cn)
**Abstract**
The straight tsunami waves from epicenter can be reflected when they reach to coasts or
underwater obstacles. In this study, we present the first ionospheric maps of reflected tsunami
signature caused by the great 11 March 2011 Tohoku earthquake using the dense GPS
network GEONET in Japan. We observed tsunami-like travelling ionospheric disturbances
(TIDs) with similar propagation characteristics in terms of waveform, horizontal velocity,
direction, period and arrival time compared to the reflected tsunami at the sea-level, indicating
the TIDs are induced by the reflected tsunami. The results confirm the atmospheric internal
gravity waves (IGWs) produced by reflected tsunami can also propagate upward to the
atmosphere and interact with the plasma at the ionospheric height.
**Keywords**: GPS; total electron content; traveling ionospheric disturbances; reflected tsunami

**1    Introduction**
A tsunami propagating in an open ocean can produce atmospheric internal gravity waves
(IGWs) and they are significantly amplified when propagate upward to the ionosphere (Hines,
1972; Peltier and Hines, 1976). The IGWs interact with the ionospheric plasma and might
generate the signatures that can be detectable by ionospheric sounding.



The detection of ionospheric signature caused by a tsunami began at 2005 on the case of the
2001 Chile earthquake, which was performed by using the ionospheric imaging derived from
very dense Japanese GPS Earth Observation Network (GEONET) (Artru et al., 2005). After
that many scholars also observed tsunami-driven TIDs in measurements of ionospheric total
electron content (TEC) derived from ground-based GPS stations (DasGupta et al., 2006;
Rolland et al., 2010; Liu et al., 2011; Galvan et al., 2011; Occhipinti et al., 2013; Tang et al.,
2015; Zhang and Tang, 2015) and satellite-based altimeters (Occhipinti et al., 2006), Doppler
sounders (Liu et al., 2006) and airglow images (Makela et al., 2011; Occhipinti et al., 2011).
Furthermore, the numerical modeling can also show evidences that ionosphere is a sensitive
medium to tsunami propagation (Occhipinti et al., 2006; Mai and Kiang, 2009; Hickey et al.,

11  2009).

Previous studies focused on the ionospheric signature induced by straight tsunami waves. For
a big size tsunami, its waves propagate to the coasts or underwater obstacles, the reflected
waves might be generated. A reflected tsunami waves were observed at the sea-level for the
May 2006 Tonga tsunami (Tang et al., 2008). Rozhnoi et al. (2014) observed a signature in
low ionosphere, which possible was generated by the reflected tsunami, following the 2010
Chile earthquake with the VLF signals. However, the VLF signals can only provide us a
single time-series with the observation time and period of the ionospheric disturbances; other
significant propagation characteristics such as horizontal velocity and direction are not
presented. Whether the IGWs induced by the reflected tsunami waves can propagate to
ionosphere is still unclear.
In this study, we will apply the two-dimensional TEC maps derived from a dense GPS
network (GEONET) to detect the possible reflected tsunami signature in ionosphere after the
2011 Tohoku earthquake. The TEC maps can image propagation of TIDs over lager areas,
which are very suitable for the ionospheric monitoring.

**2  Method**
Due to the high spatial and temporal resolution, GPS ionospheric monitoring has been a
powerful tool for remote sensing of the ionosphere. The parameter using in GPS ionospheric
monitoring is ionospheric TEC (sTEC), the integrated electron density along the entire line-





of-sight (LOS) between receiver and satellite. The sTEC can be calculated from the geometry-
free combination of GPS dual-frequency carrier phases for each satellite-receiver pair, namely
$$s_T = \frac{1}{40.3} \frac{f_1^2 f_2^2}{f_1^2 - f_2^2} \left( L_1 - L_2 + const + \varepsilon \right) \qquad (1)$$
where $s_T$ is the sTEC with unit of TECU (1 TECU=$10^{16}$/m$^2$); $f_1$ (1575.42 MHz) and
$f_2$ (1227.60 MHz) are the carrier phase frequencies, respectively, $L_1$ and $L_2$ are carrier phase
observations with unit of meter, respectively; *const* is the unknown constant bias, including
the ambiguity and instrument bias; $\varepsilon$ is the measurement noise. A single-layer model with
height of 350 km is used to obtain the vertical TEC (vTEC) $v_T(t)$ and position of ionospheric
pierce point (IPP).
Although Eq. (1) cannot acquire the absolute value of TEC at a particular time due to the
unknown bias, it can capture the TEC variation over time with high precision, which is
important for TIDs detection. In this paper, we employ a numerical difference method to
eliminate the diurnal variation and the bias in TEC and extract the vTEC variation series
(Hern ández-Pajares et al., 2006; Tang and Zhang, 2014)
$$d(t) = v_T(t) - 0.5(v_T(t-\tau) + v_T(t+\tau)) \qquad (2)$$
where $d(t)$ is the vTEC variation; $t$ is the observation epoch; $\tau$ is the time step for difference.
The time step is set to 300 s, which is suitable for the TIDs detection induced by the IGWs.
The numerical difference method is very simple and beneficial to process large number of
data.
The number of the ground-based stations in GEONET is about 1200 and data sampling rate is
30 s. The average distance between GEONET stations is about 25 km, providing us a good
opportunity to monitor the ionosphere with high resolution. After processing the data from all
GPS stations, we can plot the vTEC variation values with different IPPs at specific epoch in a
two-dimensional map. The two-dimensional maps of vTEC variation will be used to detect
the tsunami signature in ionosphere.



## 3 Results and Analysis

### 3.1 Reflected tsunami signatures at sea-level

According to U.S. Geological Survey, the Tohoku (Japan) earthquake (Mw=9) with epicenter located at 38.297 °N, 142.373 °E occurred at 05:46 UT on 11 March 2011 and then triggered powerful tsunami. Figure 1 indicates the locations of epicenter and Deep-ocean Assessments and Reporting of Tsunami (DART) bottom pressure stations operated by the National Data Buoy Center (NDBC) of U.S. National Oceanic and Atmospheric Administration (NOAA). As showing in Figure 1, when straight tsunami waves from epicenter propagate to the Emperor seamounts (average depth ~2000 km), the reflected waves might be generated due to the powerful energy.

To confirm the existence of reflected tsunami waves, Figure 2 presents the sea-level measurements recorded by the DART 21401 and DART 21419. The DART 21401 that near the epicenter first observes the straight tsunami at 06:46 UT and then the waves propagates to the DART 21419 at 07:09 UT. However, the DART 21419 and DART 21401 observes another augmented signal at 09:28 UT and at 09:52 UT, suggesting there are tsunami waves propagated along the opposite direction of the straight tsunami waves. Furthermore, the amplitude of the waves recorded in DART 21419 (0.139 m) is slightly larger than that in DART 21401 (0.111 m), which can serve as further evidence that the waves propagated from DART 21419 to DART 21401. As shown in Figure 2, the spectral analysis for the time-series of the tsunami signals indicate they have similar center frequency of ~0.40 mHz. Another significant signal (~0.66 mHz) in time-series of DART 21401 is the frequency of straight tsunami (Makela et al., 2011; Occhipinti et al., 2011).

In addition, the model of sea-level variation due to tsunami waves can further support the reflected tsunami waves. The NOAA's Method of Splitting Tsunami (MOST) model (see the animation for Tohoku tsunami: ftp://ftp.pmel.noaa.gov/tsunami/honshu/) can offer the propagation characteristics of the reflected tsunami at the sea-level. According to the MOST model, the straight tsunami waves reach the Emperor seamounts at about 3 h after the earthquake. Then, the tsunami waves are divided into two parts: one part continued to propagate ahead, another part along the opposite direction. The opposite tsunami waves pass the DART 21419 and DART 21401 successively, and reach the epicenter at about 6 h after the earthquake.



In short, the observation results by DARTs and the MOST model results demonstrate that
there were reflected tsunami waves at the sea-level with frequency about 0.40 mHz.
**3.2   Reflected tsunami signatures in ionosphere**
The ionospheric disturbances induced by origins from epicenter such as Rayleigh waves,
acoustic-gravity waves and gravity waves (tsunami waves) were observed by several scholars
after the 11 March 2011 Tohoku earthquake (Liu et al., 2011; Rolland et al., 2011; Tsugawa
et al., 2011; Occhipinti et al., 2013; Occhipinti, 2015). Here, we focus on possible reflected
tsunami signature in ionosphere induced by this event.
Figure 3 presents the two-dimensional maps of vTEC variations derived from GPS
observations in GEONET on 11 March 2011 at studied times. As shown in the upper left
panel, the ionospheric disturbances induced by origins from epicenter are basically invisible
at about 08:30 UT. However, TIDs (indicated by the blue arrow) began to appear again at
about 11:50 UT. As shown in Figure 3, the TIDs propagated along the southwest that is
accord with the observation results by the DARTs in Figure 2. In addition, the direction
slightly turns around counterclockwise, suggesting that the horizontal velocities on the
western area close to the Kuril and Japan trenches are larger than that on the eastern area. As
shown in the bottom right panel of Figure 3, the TIDs propagate to the epicenter at about
12:20 UT.
The time-distance map of vTEC variations is usually used to estimate the velocity of
disturbances with a point origin, such as the epicenter. However, the origin for the reflected
tsunami is not a point but a line (the Emperor seamounts). So, the method obtaining the
velocity of disturbances by time-distance map is not suitable. Here, we estimate the horizontal
velocities of the TIDs in vTEC variations following the approach of Garrison et al. (2007).
The horizontal velocities of the TIDs vary 240-290 m/s approximately from eastern area to
western area, which is consistent to the observed results in the ionospheric maps. According
to the ocean depths ($h$) from ETOPO1 grid data supplied by U.S. NOAA (about 5900-9000 m
at adjacent area), the tsunami velocities are about 242-300 m/s obtained from the shallow-
water equation $v = \sqrt{gh}$ with gravity ($g$) of 9.8 m/s$^2$. This indicates the horizontal velocities of
the observed TIDs are similar to the tsunami velocities at the sea-level.
To further examine the correlation of TIDs and tsunami waves, we plot vTEC variation time-
series derived from satellite PRN 25 and satellite PRN 29 in Figure 4 (The TIDs are observed



by the two satellites). Compared Figure 4 and Figure 2, it is easy to find the waveform of the
TIDs is very similar to that of the reflected tsunami waves at the sea-level. According to the
time-frequency diagrams, the center frequency of the TIDs is about 0.55 mHz, which is larger
than that of the reflected tsunami waves. This can be attributed to the Doppler Effect caused
by the relative motion between the GPS satellite and the TIDs. As seen from Figure 1, the
direction of PRN 25 or PRN 29 is almost opposite to the TIDs, leading to the shortened period
when observe the disturbances. The center frequency of TIDs is about 0.44 mHz after
amending the Doppler Effect, which is basically consistent to that of reflected tsunami waves.
According to the NOAA's MOST model for this event, the direction of the reflected tsunami
waves at the sea-level is also southwest and turns around counterclockwise, which is
consistent to that of the observed TIDs in Figure 3. As described in the previous section, the
reflected tsunami waves reach the epicenter at about 6 h after the earthquake. So, the observed
time for the TIDs and reflected tsunami is basically consistent with ~35 min delay. The
magnitude of the observed delay is also observed on the case of 2009 Samoa tsunami
(Rolland et al., 2010), which might be due to combined influence of thin shell approximation,
horizontal wind and horizontal delay when IGWs propagated to the ionosphere. The
horizontal group velocity of IGWs at different height is always smaller than the tsunami
velocity, leading to the horizontal delay when propagating to the ionosphere (Occhipinti et al.,
2013). According to the HWM93 wind model (Hedin et al, 1991), the background horizontal
winds propagate along the northwest at the studied times and areas, which are opposite to that
of tsunami waves. When IGWs propagate against the winds, the horizontal group velocity
will decrease (Ding et al., 2003), further prolonging the horizontal delay.
According to above analysis, the observed TIDs have similar horizontal velocity, direction,
period, waveform and observed time compared to the reflected tsunami waves at the sea-level.
Furthermore, to remove possible recurrent TIDs, we also verified that there weren't
significant perturbations on the day before and after the event day at the study times. These
results can confirm that the observed TIDs in Figure 3 are triggered by the reflected tsunami
waves. This is the first time that the reflected tsunami signature in ionosphere is detected by
the GPS TEC.



## 4    Conclusions

In this paper, we firstly analyze the tsunami measurements recorded by two DART buoys provided by the NDBC of U.S. NOAA after the 11 March 2011 earthquake, demonstrating that the straight tsunami waves from mainshock are reflected when propagating to the underwater obstacles. Then, we employ the two-dimensional maps of vTEC variations extracted from very dense Japanese GEONET GPS data to observe disturbances in ionosphere. The observed TIDs have similar propagation characteristics in terms of horizontal velocity, direction, waveform, period and observation time compared to the reflected tsunami waves at the sea-level, suggesting the TIDs are triggered by the reflected tsunami waves. The observed results in this study confirm the IGWs generated by reflect tsunami can also propagate upward to the atmosphere and interact with the plasma at the ionospheric height.

The results indicate not only the straight tsunami from mainshock, but also the reflected tsunami might have potential to induce ionospheric disturbances. This study provides a new recognition of tsunami-driven TIDs and support for the future tsunami warning system.

**Acknowledgments**

The GPS data used in this study are provided by the GeoSpatial Information authority (GSI) of Japan (ftp: //terras.gsi.go.jp). The DART data are provided by the National Data Buoy Center (NDBC) of U.S. National Oceanic and Atmospheric Administration (NOAA). This study was supported by National Natural Science Foundation of China (Grant No. 41231064, 41204028) and the Surveying and Mapping Foundation Research Fund Program, National Administration of Surveying, Mapping and Geoinformation (13-02-07).



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

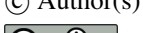



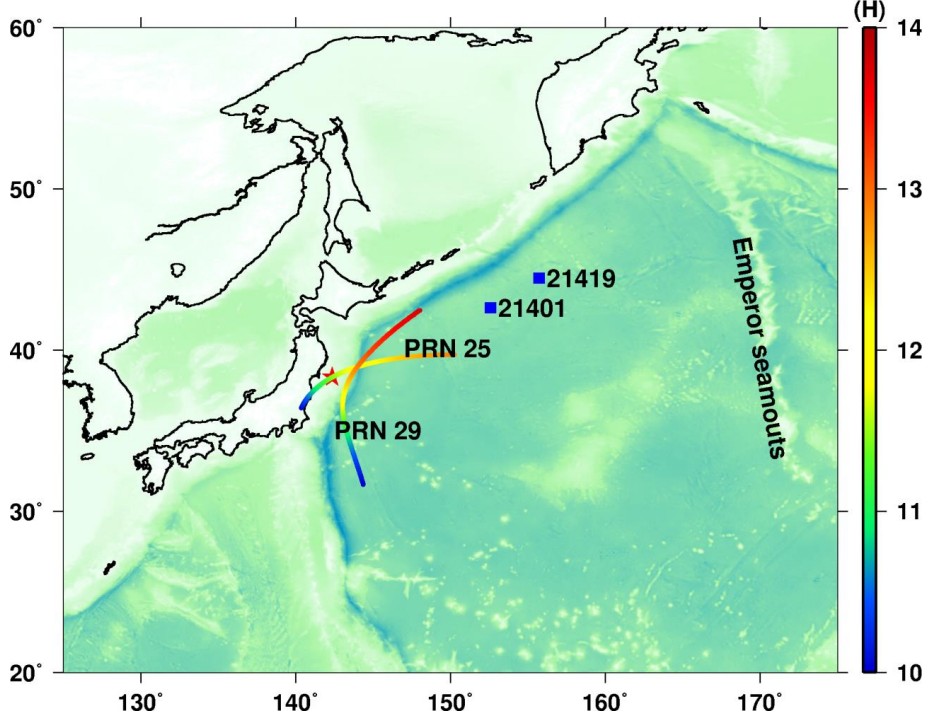

2    Figure 1. The locations of epicenter, DART stations and IPPs. The red pentagram indicate the

3    epicenter of 2011 Tohoku earthquake, the blue squares note the DART stations and the color

4    bar denotes the observation time of IPPs.




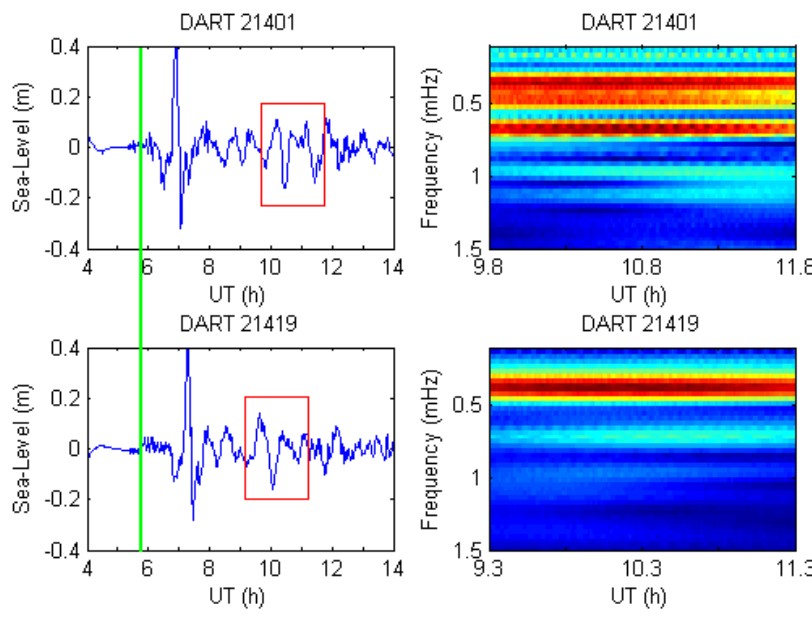

Figure 2. The sea-level tsunami series and time-frequency diagrams. The left panels are sea-level tsunami measurements recorded by the DART 21401 and DART 21419. The right panels are corresponding time-frequency diagrams for the time-series of the tsunami signals denoted by red rectangles. The green line in the left panels indicates the time of earthquake.





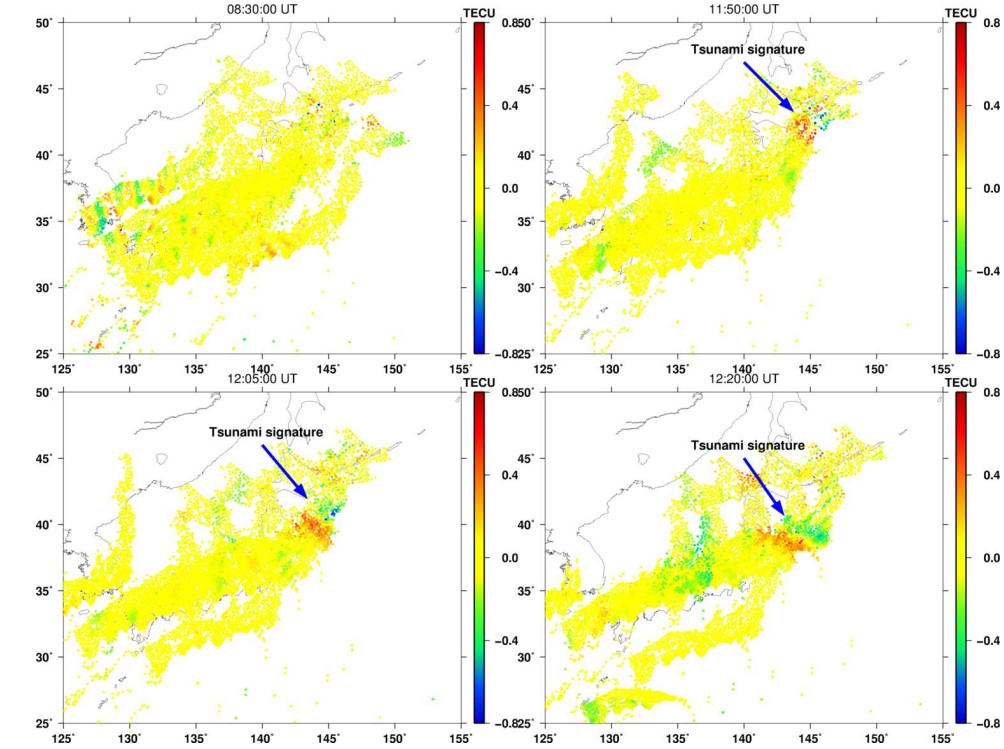

Figure 3. The two-dimensional maps of vTEC variations derived from GPS observations in
GEONET. At about 08:30 UT, the ionospheric disturbances induced by origins from epicenter
(Rayleigh wave, acoustic wave, etc.) are basically invisible. At about 11:50 UT, the
ionospheric signature driven by reflected tsunami wave is appeared.





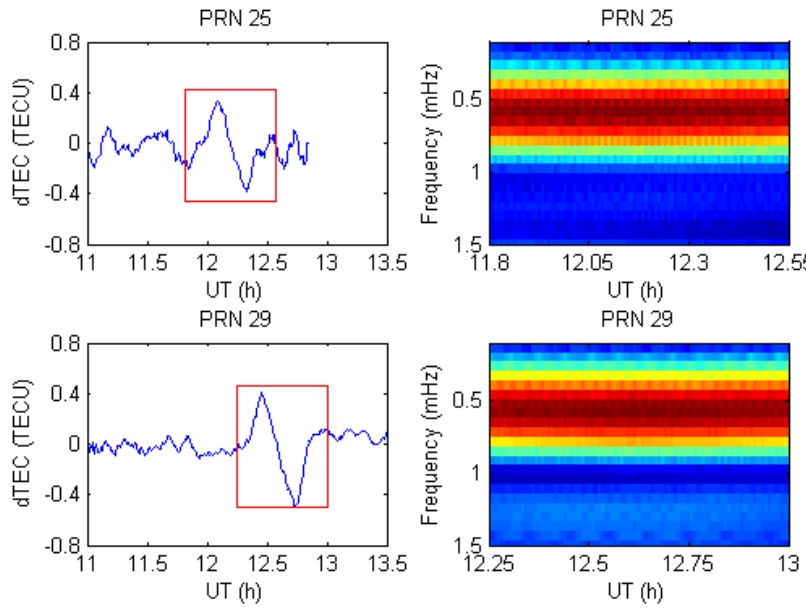

Figure 4. The vTEC variation series and time-frequency diagrams. The left panels are vTEC
variation series derived from satellite PRN 25 and satellite PRN 29. The right panels are
corresponding time-frequency diagrams for the time-series of the tsunami signals denoted by
red rectangles.

