# Peer review of "First GPS TEC maps of ionospheric disturbances induced"

_Natural Hazards and Earth System Sciences, 2016_

## Referee Comment (RC1) · Anonymous Referee #1 · 17 Feb 2016

Comments on the manuscript "First GPS TEC maps of ionospheric disturbances induced by reflected tsunami waves: The Tohoku case study" by Long Tang et al.

This manuscript presents a clear depiction of the signatures generated by the Tohoku tsunami in GPS TEC. Accumulating observational evidences of tsunami-induced atmospheric gravity waves is important for better understanding the coupling mechanisms between the ocean and the atmosphere. In this study, the reflected tsunami signatures are observed both in ionosphere and at sea level, suggesting the results are reliable. The paper is worth of publication after appropriate revisions.

1. Page 3, line 3. Undoubtedly, the Eq. (1) is right. However, the unit for the constant 40.3 should also be indicated, which is m3/s2. Then, the unit is consistent (TECU or

m-2) for both sides of the Eq. (1).

2. Page 3, line 7-9. Some references that describe the single-layer model and iono-spheric delay estimation should be included. For example: Li, X., M. Ge, H. Zhang, and J. Wickert (2013), A method for improving uncalibrated phase delay estimation and ambiguity-fiAxing in real-time precise point positioning, J. Geod., 87(5), 405-416, doi:10.1007/s00190-013-0611-x.

3. Page 5, line 21. "...tsunami is not a point but a line (the Emperor seamounts)". The location for the Emperor seamounts is illustrated in Figure 1, but should also be indicated in text. In the introduction, you should mention a little bit about earth-quake/tsunami monitoring and early warning using GNSS coseismic displacements, e.g., doi:10.1002/grl.50138, and doi: 10.1093/gji/ggt249

4. Page 5, line 22-23. Although the reference is provided, the authors need to simply explain the method how to estimate the TID propagation characteristics.

5. Page 7, line 2. "...we firstly analyze the tsunami measurements...", "firstly" should be "first".

6. In the conclusion section, please also discuss and provide some outlook about multi-GNSS, which is the future of GNSS development. FYI, Li, X., M. Ge, X. Dai, X. Ren, M. Fritsche, J. Wickert, and H. Schuh (2015), Accuracy and reliability of multi-GNSS real-time precise positioning: GPS, GLONASS, BeiDou, and Galileo, J Geod., 89, 607–635, doi: 10.1007/s00190-015-0802-8. Li, X., F. Zus, C. Lu, G. Dick, T. Ning, M. Ge, J. Wickert, and H. Schuh (2015), Retrieving of atmospheric parameters from multi-GNSS in real time: Validation with water vapor radiometer and numerical weather model. J. Geophys. Res. Atmos., 120, 7189–7204. doi: 10.1002/2015JD023454.

7. Please try your best to cite peer-reviewed journal papers and to avoid conference papers or reports.

Best regards,

[Figure]

Please also note the supplement to this comment:
http://www.nat-hazards-earth-syst-sci-discuss.net/nhess-2016-11/nhess-2016-11-RC1-supplement.pdf

————————————————————

---

## Referee Comment (RC2) · Anonymous Referee #2 · 19 Apr 2016

Comments to the Author:

Review for *Natural Hasards and Earth System Sciences* manuscript NHESS-2015-11 « First GPS TEC maps of ionospheric disturbances induced by reflected tsunami waves », by L. Tang *et al.*

The paper submitted by Tang *et al.* presents the detection of the reflected tsunami signature in the ionosphere following the Tohoku event. Authors highlight the originality of their work claiming, as resumed in the title, the « first GPS TEC maps of ionospheric disturbances induced by reflected tsunami waves ». This is literally true, but I wish to highlight several points in order to clearly define the weight of this first detection:

1) The GPS data and the TEC map showed by Tang *et al.* was already published by several authors (*e.g.*, Rolland et al. 2011, Chen et al., 2011, Tsugawa et al., 2011, Liu et al., 2011, Maruyama et al., 2011, Saito et al., 2011) just after the Tohoku event (special EPS with submission within the first month after the event).

2) The reflected tsunami is stil a tsunami, and the reflection usually don't change radically the properties of the tsunami wave. Consequently if the tsunami produces an atmospheric internal gravity wave perturbing the plasma, this is also the behavior expected for the reflected tsunami. The coupling phenomena between the ocean, the atmosphere and the ionosphere, is clearly understood, measured and reproduced by modeling from the Sumatra tsunami (Occhipinti et al., 2006, 2008, 2011, 2013, Occhipinti 2015).

3) Tang *et al.* focalize their attention on the TEC perturbation induced by the reflected tsunami.

The paper deserves for publication in *Natural Hasards and Earth System Sciences* as it's interesting to focalize on the signature of the reflected tsunami, but I feel that authors put an exaggerate emphasis on the explorative value of their work. This attitude reduce the interest of their work as it gives the impression that they don't deeply know the literature on the tsunami detection by ionospheric sounding.

Authors can find here some comments & suggestions to improve the interest of this work.

The most important comment concern the comparison between the theoretical and observed arrival time of the reflected-tsunami signaure in the ionosphere.

**Comments & Suggestions:**

**Page 1, lines 27-28:** « The IGWs interact with the ionospheric plasma and might generate the signatures that can be detectable by ionospheric sounding ». Please, use « generate » instead of « might generate », the coupling between the tsunamis and the ionosphere is fully proved by observations and modeling (Occhipinti et al., 2006, 2011), generalized for several events (Rolland et al., 2010, Occhipinti et al., 2013).

**P. 2, l. 1-11:** Authors resumes in this ten lines the entire review of the tsunami detection by ionospheric sounding. The use of the references is correct, but authors could maybe extend the description in order to general neophyte readers. Please, see Occhipinti (2015) for a recent complete review.

**P. 2, l. 20-21:** « Whether the IGWs induced by the reflected tsunami waves can propagate to inosphere is still unclear ». Please, change with « Even if theoretically detectable, the ionospheric signature of the IGWs induced by the reflected tsunami is not yet supported by observations ».

**P. 3, l. 10-19:** The method used by the authors correspond to a low-pass filter taking the signal with periods longer of 5 min, please highlight it in the main text. I also strongly suggest to compair their technique with another filter (e.g., butter filter) applied to Vt and show the two results in Figure 4.

**P. 4, l. 16-19:** The fact that the amplitude observed at DART 21419 is langer than DART 21401 is not a proof of the presence of reflected tsunami. Indeed, the amplitude strongly depends by the bathymetry. In order to prove that the observed wave is the reflected tsunami, please, compare with a tsunami numerical modeling. You can show the DART observations, and comparison with the tsunami modeling, highlighting the direct and reflected tsunami.

**P. 5, l. 9-29:** The discussion about the speed and the arrival time needs to be supported by the coupling theory between ocean/atmosphere/ionosphere in order to validate the arrival time of IGWs. Please, refer to Occhipinti et al. (2013, fig.7, 8 and 9) to compute the arrival time and the distance from the coast of a visible IGW in the ionosphere related to the tsunami. Indeed, the reflected tsunami is generated at the coast, consequently it is visible in the ionosphere one hour after and 500km from the coast where it is generated. Please, note that this values (1h and 500km) are approximative as the speed depends of the tsunami period, authors can find a more exact valus in Occhipinti et al. (2013) related to the observed period of the reflected tsunami. This is the most important comment of my review, and I think that supporting their result with the theory, the author could straggly improve the impact of their observational results and can also clarify the discussion about the delay presented on **P.6, l.12-16**.

**P. 6, l. 1-5:** First, please, show in Figure 4 the reflected tsunami wave observed by the DART (use two different y-axis on right and left for dTEC and Sea-level). Second, the different main frequency between the DART and the dTEC it could be related to your filter method. Please, as suggest above, try also another filtering approach.